

# The Global Streamflow Indices and Metadata Archive (GSIM) – Part 1: The production of daily streamflow archive and metadata

Hong Xuan Do[1], Lukas Gudmundsson[2], Michael Leonard[1], Seth Westra[1], Sonia Isabelle Seneviratne[2]

[1]School of Civil, Environmental and Mining Engineering, University of Adelaide, Adelaide, Australia
[2]ETH Zürich, Institute for Atmospheric and Climate Science, Zürich, Switzerland

*Correspondence to*: Hong Xuan Do (hong.do@adelaide.edu.au)

**Abstract.**

This is the first part of a two paper series presenting the Global Streamflow Indices and Metadata archive (GSIM), a worldwide collection of metadata and indices derived from more than 35,002 daily streamflow timeseries. This paper focuses on the compilation of the daily streamflow timeseries based on 12 free-to-access streamflow databases (seven national databases and five international collections). It also describes the development of three metadata products (freely available at https://iacweb.ethz.ch/staff/lukasgu/GSIM/GSIM_metadata.zip): (1) a GSIM catalogue collating basic metadata associated with each timeseries, (2) catchment boundaries for the contributing area of each gauge, and (3) catchment metadata extracted from 12 gridded global data products representing essential properties such as land cover type, soil type, climate and topographic characteristics. The second paper in the series then explores production and analysis of streamflow indices. Having collated an unprecedented number of stations and associated metadata, GSIM can be used to advance large-scale hydrological research and improve understanding of the global water cycle.

## 1 Introduction

Having streamflow observations with global coverage is essential for progressing the science of large-scale hydrology. For example, Global datasets provide particularly value when evaluating global hydrological models (Gudmundsson et al., 2012; Huang et al., 2016; Ward et al., 2013), investigating large-scale weather patterns and their relation to hydrological extremes (Wanders and Wada, 2015; Ward et al., 2014), and detecting changes in global hydrological extremes over space and time (Do et al., 2017; Kundzewicz et al., 2012; Milly et al., 2002), amongst numerous other applications.

Despite the fundamental, widespread and varied applications that streamflow observations support, there are many obstacles to the existence and utility of a large-scale streamflow archive. Firstly, there are threats to the quantity of data, such as, political sensitivities (Nelson, 2009), cost recovery and strict access policies (Hannah et al., 2011), unavailability in an electronic format, consistency of data formats, limited documentation, missing metadata, and a lack of agency for database maintenance and updating. Secondly, there are difficulties associated with the quality of the data in many regions, such as poor spatial coverage, poor quality control, variable quality control between regions, inconsistent metadata, imprecise geographic coordinates of the site, changes in the density of stream gauges and variable record lengths. Lastly, even in locations where there are abundant and high quality streamflow observations, there can be questions over its utility due to the manifestation of human impacts—for example, urbanization, land-use changes, channelization and upstream dams (Hannah et al., 2011).

To date, the Global Runoff Data Centre database (GRDC) has been the primary dataset used in large-scale hydrological studies, with more than 9000 stations available to the research community (GRDC, 2015).The GRDC database is supported on a voluntary basis and the number of data submissions depends on contributions by national authorities. However, although numerous countries have databases of acceptable quality, they cannot afford the resources to supply



their data and the GRDC remains sparse in some regions. For example, the catalogue of the GRDC database (version May

12th 2016) shows that from 7,092 daily timeseries, there are only 630 stations over South America and only 643 stations

over Asia. Moreover, many stations in regions such as Asia and Russia have not been updated for many years and are

missing otherwise available data at the end of their records.

The Global Streamflow Indices and Metadata (GSIM) project has been initiated in order to address the demand for a

global streamflow database (Bierkens, 2015; Fekete et al., 2015; Hannah et al., 2011; Kundzewicz et al., 2013; Merz et

al., 2012; Milly et al., 2015). The activities of the project have been to collate publicly available data, apply basic

consistency to the formatting and establish a standardised set of metadata. In so doing, GSIM intends to promote more

widespread use of streamflow data, facilitate improved research outcomes through increased spatial coverage and gauge

density, and tackle ongoing challenges for the hydrological community; for example, addressing fundamental issues of

data quality, identifying additional data sources, lobbying for continuity of data networks and developing a method for

improved governance and maintenance of streamflow data at the global scale.

To maximise the value of the streamflow dataset for a range of applications, the GSIM project also seeks to provide

information on catchment characteristics upstream of the streamflow gauging station. This necessitates a consistent

approach to delineating the upstream catchment boundary for every gauge station, and this is achieved using data from a

global Digital Elevation Model (DEM). This is because, with the exception of the GRDC databases, catchment boundaries

representing the direct drainage area of stations were unavailable. Filling in this missing element of metadata is important

to facilitate further analysis of the streamflow observations with respect to a wide and ever-increasing variety of spatial

datasets. Although there have been previous efforts in producing catchment boundaries for a smaller number of stations

(Addor et al., 2017; Arsenault et al., 2016; Lehner, 2012; Schaake et al., 2006), similar work at this magnitude has not

been undertaken. This task is complicated by a lack of precision in the supplied geographic coordinates of a given site;

for example when a catchment boundary is extracted, the corresponding area may not match the reported area of the

catchment and a procedure for checking minor shifts in the coordinates is needed to improve identification of the likely

catchment boundary.

The availability of catchment boundaries for each gauge enables the association of environmental variables to each gauge

by extracting them from corresponding global-scale gridded products. As part of the GSIM project, a number of global

data products are provided as an additional dataset so that a user can readily filter the GSIM dataset according to specific

interests; for example climate type, soil type, land-use type, irrigation area and population density. Other potential

applications of this auxiliary information might include a comparison to a database of dams for identifying upstream

impacts, to remotely sensed estimates of forest cover or urban extent for determining land-use change, to population

demographics for improving estimates of flood exposure, and to hydrological model outputs for evaluating model

performance.

Finally, to facilitate benefits of this project to the broader community, indices characterising water-balance aspects,

hydrological extremes and features of the seasonal cycle have been derived from the GSIM timeseries and will be made

publicly available. To ensure standardised quality for the derived indices, a quality control procedure coupling the

information provided by data providers and a data-driven approach was also applied.

This is the first paper of a two-part series detailing the production of GSIM and corresponding data-products. This paper

outlines the provenance of daily streamflow timeseries (Section 2), procedures for reformatting and combining the

timeseries (Section 3), the development of metadata associated with each gauge (Section 4), data availability (Section 5)

and an overall summary of the GSIM timeseries and metadata (Section 6). The second paper in this series (Gudmundsson

et al., submitted) provides significantly more detailed analysis of the GSIM data, including: (1) checks for data quality,

(2) the production of streamflow timeseries indices, and (3) homogeneity assessment of the derived indices.



## 2 Daily streamflow data and where to find them

GSIM is a compilation of 12 databases having either open-access or restricted-access policies, giving a total of 35,002 stations. The spatial distribution and the number of stations available in each database is illustrated in Fig. 1. A summary of the data sources is also provided in Table 1 and detailed information on each database is elaborated upon in following sections. The list of databases identified as part of GSIM is not exhaustive of all possible data sources, only of those that were known to the authors and readily accessible within the project time frame. Where additional data is available in a convenient format it is possible to further augment GSIM in the future.

The various data sources were classified as either a "research database" or a "national database". The reasons for this classification are further outlined in Section 3, but relate to issues when merging databases and removing duplicate gauges. The data sources include:

(1) **Research databases:** Databases with daily streamflow data that have been compiled on an ad hoc basis from a variety of original sources by research organisations. This category includes five different databases: The Global Runoff Data Centre database (GRDC); the European Water Archive (EWA); the China Hydrological Data Project (CHDP) data archive; the GEWEX Asian Monsoon Experiment – Tropics (GAME) data archive; and the Regional Hydrographic Data Network for the Arctic Region (ARCTICNET) data archive.

(2) **National databases:** Databases with daily streamflow data made publicly available by national water authorities as part of water-related regulations. This category includes seven databases: The USGS Water Data for USA database (USGS); Canada's National water data archive (HYDAT); Japan's Ministry of Land and Infrastructure database Water Information System (MLIT); Spain's digital hydrological year book database (Anuario de aforos digital 2010–2011, AFD); Australia's Bureau of Meteorological Water Data Online database (BOM); India's Water Resources Information System database (WRIS); and Brazil's National Water Agency database (ANA).

### 2.1 The Global Runoff Data Centre (GRDC) dataset

The daily streamflow dataset received from the Global Runoff Data Centre (6,313 stations with greater than 10 years record, see also Gudmundsson and Seneviratne (2016)) is referred to as the GRDC in this project. To date, the GRDC dataset has been the largest and most extensively used dataset for streamflow analysis at regional and global scales. It was thus considered as the starting point and "base" for the GSIM project. Indeed, it was awareness of data "missing" from the GRDC that prompted the initial search for additional sources of data to complement the database.

The GRDC was initiated in 1988 by the World Meteorological Organisation and is now maintained at the German Federal Institute of Hydrology in Koblenz. Although the GRDC provides free and unrestricted access to all hydrological data and products, the data policy indicates that requests for data must reach the GRDC in written form to ensure data users do not redistribute the timeseries. More detail about the GRDC data policy, and procedure for obtaining its timeseries, are outlined at http://www.bafg.de/GRDC/EN/01_GRDC/12_plcy/data_policy_node.html

### 2.2 The European Water Archive

The European Water Archive (referred to as the EWA in this paper) is one of the most comprehensive streamflow timeseries archives in Europe, with more than 3000 river gauging stations distributed over 29 countries. This archive is also currently held by the GRDC, but the EWA data policy indicates that the archive is available free of charge only to EURO-FRIEND project members with permission from the relevant EURO-FRIEND project coordinator (http://ne-friend.bafg.de/servlet/is/7413, last access: 23 June 2017). The EWA stations used in this paper were selected using the same criteria as Gudmundsson and Seneviratne (2016), with a total number of 3,731 daily records.



### 2.3 The China Hydrology Data project


The China Hydrology Data Project (referred to as the CHDP in this paper) aims to digitise an arrangement of hydrological measurements taken at Chinese stations until 1987. These measurements (including daily discharges) were originally only available in book form (Henck et al., 2010). The original data were collected by the Chinese Hydrology Bureau and published in annual yearbooks. At the time GSIM began, discharge data were only available for the Yunnan-Tibet

International Rivers, which corresponded to 163 stations. The data and metadata were obtained directly from the author of the project. Detailed information can be viewed at http://www.oberlin.edu/faculty/aschmidt/chdp/index.html (last access: 23 June 2017).

### 2.4 The GEWEX Asian Monsoon Experiment – Tropics project

The GEWEX Asian Monsoon Experiment – Tropics project (referred to in this paper as GAME) was initiated in 1996 to

monitor several hydro-climatological variables over the humid temperate area in south-east Asia. As one of several important activities in this project, many hydrological observation datasets were collected, including streamflow data. Available streamflow data was provided by the Royal Irrigation Department of Thailand (RID), and comprised 129 timeseries spanning a period from 1980 to 2000. Daily discharge data and associated metadata were archived and can be accessed online at http://hydro.iis.u-tokyo.ac.jp/GAME-T/GAIN-T/routine/rid-river/index.html (last access: 23 June

135 2017).

### 2.5 The ARCTICNET project

A regional, hydrometeorological data network for the pan-Arctic Region project is a regional data bank that can be accessed via the internet and is referred to as ARCTICNET in this paper. The databank is designed to support hydrological sciences and water resource assessments over this region with the goal of estimating the contemporary water and

constituent balances for the pan-Arctic drainage system. Although most data provided in the data portal are at monthly resolution, there are 139 high-quality, daily streamflow timeseries across Russia that are also available. These timeseries, along with their metadata, were archived and can be downloaded at http://www.r-arcticnet.sr.unh.edu/v4.0/index.html (last access: 23 June 2017).

### 2.6 The USGS database

The USGS National Data Services for the US provide access to water resources data collected at approximately 1.5 million sites in all 50 States of the USA, also including the District of Columbia, Puerto Rico, the Virgin Islands, Guam, American Samoa and the Commonwealth of the Northern Mariana Islands. All timeseries and associated metadata can be queried from the data portal http://waterdata.usgs.gov/nwis (last access: 23 June 2017). To ensure the queried data have sufficient geographic metadata (critical for the present project), the stations listed in the "Geospatial Attributes of

Gages for Evaluating Streamflow, version II" (GAGES II) database were used (Falcone, 2011). The timeseries from 9,404 stream gauges obtained from the USGS data portal are referred to as the USGS database in this paper.

### 2.7 The HYDAT database

Canada's National Water Data Archive (HYDAT) is a database containing daily observed hydrometric data from publicly funded gauges in Canada. Also available in the HYDAT database are metadata about the hydrometric stations, such as

latitude and longitude, catchment area, record length, as well as information regarding flow conditions (current status, regulated or natural regime). The database is updated four times per year and currently contains data for 6,325 streamflow



stations across Canada. The raw data, as well as an extractor executable, are publicly available from Environment Canada's website at https://ec.gc.ca/rhc-wsc/default.asp?lang=En&n=9018B5EC-1 (last access: 23 June 2017).

### 2.8 The ANA database

The data portal HIDROWEB was organised by the Brazilian National Water Agency (ANA). It provides a database with all the information collected by Brazil's hydrometeorological network. Streamflow data and associated metadata were made publicly available by Brazil's national water regulations, and have been used extensively to monitor critical events, such as floods and droughts. Individual timeseries and their associated metadata can be viewed or downloaded at http://hidroweb.ana.gov.br (last access: 23 June 2017). The 3,313 stations downloaded from this website are referred to

as the ANA in this paper.

### 2.9 The AFD database

Spanish streamflow data were retrieved from the digital hydrological year book (Anuario de aforos digital 2010–2011, AFD), which provides observations until 2013–2014 and is freely accessible online (http://ceh-flumen64.cedex.es/anuarioaforos/default.asp (last access: 23 June 2017). For the GSIM, we recycled the timeseries that

was used to develop the E-RUN dataset (Gudmundsson and Seneviratne, 2016). The original DVD containing the full database was obtained directly from the Spanish authorities via a written form request. This collection contains streamflow data from 1,197 gauging stations, and is referred to as ADF in this paper.

### 2.10 The MLIT database

In Japan, the Ministry of Land, Infrastructure, Transport and Tourism is responsible for organising hydrological data. All

records are disseminated at http://www1.river.go.jp/ (last access: 23 June 2017). As at 2010, the database kept records of all river stations (at both discharge and gauge level). The composition of the 15-digit station IDs is outlined in the file http://www1.river.go.jp/kitei_sosoku.pdf (PDF), and can be used to query and download timeseries, along with its metadata. As the whole database is recorded in Japanese, the package 'translateR' (Lucas and Tingley, 2016) was used to translate the metadata into English. The timeseries downloaded from the Japanese water data portal (1,029 stations in

total) is referred to as MLIT in this paper.

### 2.11 The BOM database

As part of the water reform program established in Australia, Water Data Online was created to provide free access to nationally consistent, current and historical water information. It can be accessed at http://www.bom.gov.au/waterdata (last access: 23 June 2017). Water Data Online also contains historical data from some stations that are no longer

operational. Users can view or download individual streamflow timeseries from the data portal, along with standardised data and reports. The timeseries measured at 2,941 stations obtained from Water Data Online is referred to as the BOM database in this project.

### 2.12 The WRIS database

The "Generation of Database and Implementation of Web Enabled Water Resources Information System in the Country"

project (India-WRIS WebGIS) was initiated as a joint venture of the Indian Central Water Commission (CWC) and the Indian Space Research Organization (ISRO). Unclassified data can be accessed online and free of charge at: http://www.india-wris.nrsc.gov.in/wris.html (last access: 23 June 2017), while the metadata is documented at:



http://www.cwc.nic.in/main/downloads/Hydrological%20network%20details%20of%20CWC.pdf. All 318 stations were downloaded from the website. They are referred to as the WRIS database in this paper.


The production of timeseries and metadata for GSIM comprises several stages due to the range of data formats and significant variation in the quality of metadata across data sources. To ensure GSIM is presented in a transparent manner, the following sections outline procedures that are used to collate the timeseries across (Section 3), and to produce the metadata (Section 4).

## 3 Procedure for combining databases


Several of the identified data sources share common spatial domains, where typically the research databases may contain selected gauges from the national databases. It is therefore important to correctly identify duplicate timeseries when merging the databases. To maximise the quality of combined timeseries and minimise the requirement to combine timeseries, this task is conducted following three sequential steps: Step 1 – pre-processing the data to a common structure;

Step 2 – replacing all GRDC stations in countries that have a national database; and Step 3 – identifying remaining duplicates. From the 35,002 gauges, 3,197 (2,958 and 239 gauges from GRDC and EWA databases respectively) were replaced by national databases in Step 2, and 846 cases of 'very likely identical' stations were identified and removed in Step 3, leaving 30,959 'duplication-free' timeseries available in the GSIM.

### 3.1 Pre-processing the timeseries into a singular data structure

One of the major challenges in producing consistent streamflow indices is that data from different sources have different structures and storage formats. For example, the MLIT database divides streamflow records at one location into separate text files, and each file contains streamflow measurements for one year. In comparison, the HYDAT archive includes streamflow measurements from all available stations in a single matrix.

To address the varying standards of data management, the first step in combining the databases was to reformat all the

streamflow records to ensure that each timeseries is kept in a consistent format. Using the GRDC as a guide, it was decided to store all data for a given site in a single text file with three columns: a) date of measurement, b) value of measurement and c) original quality flags (if available), and with basic metadata (e.g. station name, ID, etc.) stored in the header of the file. All additionally derived metadata (i.e. from global gridded products) is stored in the station catalogue. The streamflow measurements were also converted into consistent units (cubic meters per second).

Metadata that have special characters in foreign language sources were also pre-processed into ASCII encoding system. For river names and station names that are recorded in Spanish (ADF) or Portuguese (ANA), the special characters were replaced by plain alphabetic characters using the core function iconv() of the R programming language. For river names and station names that are recorded Japanese characters (MLIT), The R package 'translateR' (Lucas and Tingley, 2016) was used with the Google Translate API for this task. Although there are some limitations related to this toolset (e.g.

some Japanese characters remaining untranslated and requiring manual translation; inconsistency in the translated results using the same original Japanese characters), this option was chosen to enable an automated and expedient translation. As a result, any text-related metadata associated with Japanese stations should be treated with care.

### 3.2 Replace the GRDC stations with national databases, if applicable

The GRDC streamflow records are themselves originally provided by national water agencies. Therefore, it seemed

reasonable to replace GRDC stations for countries where an equivalent national database was available. While this



approach is efficient, there is a potential downside of removing GRDC stations that were not otherwise present in the national data depositories, perhaps due to differences in maintenance of the databases. Nonetheless, this decision seems reasonable since the number of stations available in the GRDC is much lower than that available in national databases for all countries (see Table 2). As a result of this step, 2,958 stations located in seven countries (Australia, Brazil, Canada,

India, Japan, Spain, and the United States) were removed from the GRDC collection. In addition, 239 stations located in Spain were also removed from the EWA archive.

**3.3 Identify and remove duplicates in research databases**

The method of de-duplicating timeseries involves identification of duplicates where two data sources have overlapping coverage and potential merging of two records at a duplicated site to create a unified record. The de-duplication step was

generally undertaken between the GRDC and a 'paired' dataset (e.g. GRDC and GAME). The only exceptions for this step are for GRDC, EWA and ARCTICNET, as these three datasets share Russia as a common spatial domain.

The techniques adopted for combining research databases were based on the de-duplication procedures developed in Gudmundsson and Seneviratne (2016), which consists of three sequential steps:

(1) **Identification of 'duplication-candidates' using metadata similarity.** This step aims to identify timeseries

with a high level of similarity in metadata (either within one database or across different databases). We used three similarity metrics to identify potential timeseries: (1) Jaro–Winkler distances, a metric representing the alphanumeric similarity of strings (Christen, 2012), applied to river names of two records; (2) Jaro–Winkler distances between station names of two records; and (3) geographical proximity estimated from geographical coordinates between two records. These metrics were normalised to have the same range between 0 and 1,

where a value of 0 indicates identical metadata (e.g. the same geographic coordinates). This similarity analysis was run for each pair in the pool of stations, and any pair with an average value below 0.25 was identified as candidate duplicate records.

(2) **Classifications of duplication-candidates using timeseries similarity.** This step aims to decide whether a specific pair of duplication-candidates is likely to be identical. The overlapping period and correlation

coefficient were used as criteria for making a decision. Firstly, all duplication-candidates that do not share any overlap in their period of record are kept in the final GSIM collection, as they can represent separate timeseries even if they measured discharge at the same geographical location (e.g. due to reconstruction of the gauging station). Secondly, any timeseries with a correlation coefficient ($R^2$) lower than 0.90 was automatically identified as 'very likely different' (26 pairs), whereas $R^2 > 0.99$ indicates 'very likely identical' timeseries

(786 pairs). Finally, candidates with $0.90 \leq R^2 \leq 0.99$ (65 pairs) were visually inspected and manually classified as 'very likely identical' (60 pairs) or 'very likely different' (five pairs). All timeseries in the 'very likely different' category were retained while stations of the 'very likely identical' category were processed using the de-duplication procedure (see below).

(3) **De-duplication of identical timeseries:** regardless identical timeseries come from either the same database or

from different databases, records with the greater number of data points in the streamflow timeseries were kept while the other(s) were discarded. Although this approach has the downside of truncating the length of useful records, the number of timeseries that could be influenced by this approach is relatively low (846 timeseries, corresponding to 2.7% of the total number of available streamflow records).

A visual example of the de-duplication procedure is provided in Fig. 2. The left panel demonstrates a case of 'very likely

identical' stations, when station number 2964035 in the GRDC database was identified as an identical gauge to W.16 in





the GAME archive, based on the similarities between the provided metadata and correlation coefficient. The timeseries representing station 'GAME_W.16' was kept in the final collection, while timeseries 'GRDC_2964035' was removed. The right panel in Fig. 2 demonstrates a case of 'duplication-candidates' with correlation coefficient of 0.92 (timeseries 'GRDC_6123645' and 'EWA_9110028'). These timeseries were visually inspected, assigned 'very likely different' label, and both timeseries were kept in the final collection.

## 4 Production of the GSIM metadata

Providing a consistent set of metadata for each site has been a significant undertaking for GSIM. This section outlines three main stages to developing the GSIM metadata: (1) consolidating all available basic metadata, (2) consistently delineating catchment boundaries for each site, and (3) develop a supplementary set of catchment-scale metadata based on delineated boundaries.

### 4.1 Consolidating basic metadata from available sources

Following the GRDC format, each timeseries was accompanied by basic metadata, including:

(1) station ID
(2) station name
(3) river name of gauging location
(4) geographical coordinates of station
(5) elevation of station
(6) drainage area
(7) catchment boundary form original data sources.

This data is useful for filtering stations according to specific criteria and purposes of the analysis. Moreover, the availability of a catchment boundary for the gauge enables additional catchment-scale metadata to be derived as necessary. However, not all of this basic metadata was available for all data sources. For example, the catchment boundary was only available for parts of the GRDC and EWA stations, the drainage area was unavailable in the BOM and MLIT databases, and though several data sources included river names in station names (BOM, HYDAT, USGS), this metadata was unavailable in English for other sources (MLIT, ANA, ADF). Table 3 further outlines availability of basic metadata for each source.

The method for consolidating basic metadata for each station follows three steps:

**Step 1. Transfer and review metadata available from original sources**.

The transfer of all existing metadata required a range of simple consistency checks and conforming rules, including:

(1) Reviewing geographical coordinates of all stations. Stations with unreasonable locations (e.g. located in the middle of North Atlantic Ocean without any land mass, identified from Google Earth) were marked to be excluded from the subsequent delineation procedure (24 stations).

(2) Separating the river name from the station name. Several sources use a consistent format for the station name consisting of two parts: The name of the station followed by the name of the water body. This pattern used a formula with 'linking words' such as 'at', 'upstream' and 'downstream'. Taking station 'BOM_406219' with original station name 'Campaspe River at Lake Eppalock (Head Gauge)' as an example, the position of linking word 'at' was identified and used to extract 'river' metadata (Campaspe River) from the full station name.



(3) Retaining the metadata of duplicated timeseries having the most data points in contrast to the other timeseries being removed. While this step may mistakenly remove some information, it is expedient and reflects the typical result of de-duplicated records that longer timeseries were kept while the shorter timeseries were removed.

**Step 2. Generate 'database-merging' information**

This step documents a summary of efforts taken in creating consistent set of GSIM metadata and allows a user to check steps that were taken or to identify better procedures using alternative timeseries or metadata obtained from original sources. There are 12 fields documented for this purpose, including:

(1) an indication of whether the timeseries de-duplication procedure was used (one field)

(2) which database and station was kept to construct the GSIM timeseries (two fields)

(3) which station that were removed and the corresponding database (three fields)

(4) the value of metrics that represent similarities in the timeseries metadata (five fields)

(5) the number of overlapping days, if applicable (one field)

**Step 3. Generate information about data availability**

The last step in compiling basic metadata for GSIM was to generate metrics that represent data availability for each GSIM timeseries, including the temporal coverage (i.e. the first and final year), the number of available daily observations, the number of missing data points, and the proportion of missing data points.

**4.2 Catchment delineation procedure**

Although catchment boundaries were readily available for parts of the GRDC and the EWA databases, they were not available for other data sources. With the ever-increasing availability of remote-sensing and modelled data products at global and continental scales, the provision of catchment boundaries is an important mechanism for extending the utility of GSIM. Although catchment boundaries can be generated easily using standard delineation algorithms in GIS packages, it requires a global coverage DEM dataset and reliable locations to represent the outlet of each drainage area, which were unfortunately not readily available for GSIM project. For DEM dataset, HydroSHEDS dataset was the most reliable dataset for this purpose but it does not cover regions above 60 degrees northern latitude. To overcome this hurdle, an additional DEM dataset was used to fill up HydroSHEDS's white-space. Other challenges in this procedure are possible errors in the geographical coordinates represent the catchment outlet that can lead to unreliable result of the delineation procedure such as typos in reported coordinates (e.g. 13.47N instead of 14.47N), swapped order of the coordinate digits (e.g. 103.45E instead of 103.54E). As a result, an algorithm to identify location that well represents catchment outlet was also applied. This section describes the DEM products used, and the algorithm to identify the "best outlet" associated with each station in GSIM project.

As there were more than 30,000 stations needing to be delineated, the HydroBASINS dataset was used, dividing the world into 24 regions, so that the task of delineation could be performed in parallel. The regions are shown in Fig. 3 and are generally independent in terms of drainage areas (Lehner and Grill, 2013). North America and Europe were specifically broken into more regions to address their relatively higher density of gauges.

The main DEM product used for GSIM was HydroSHEDS (http://hydrosheds.org), which is available at 15 arc-second resolutions (Lehner et al., 2006), and has been used extensively in large-scale hydrological studies (Do et al., 2017; Lehner and Grill, 2013; Lehner et al., 2008; Wood et al., 2011). To address a limitation in the coverage of HydroSHEDS (regions above 60 degrees North, and some islands), the Viewfinder Panoramas elevation product at 15 arc-second resolutions was used (http://viewfinderpanoramas.org, last access: 25 June 2017). This dataset has been used in several studies as an alternative DEM product to overcome similar data coverage issues (Barr and Clark, 2012; Fredin et al., 2012; Sil and




Sitharam, 2016; Yamazaki et al., 2015). The quality of the Viewfinder Panoramas is not as clearly documented as for HydroSHEDS, thus its use was kept to a minimum. To maintain consistency when delineating boundaries only one DEM product was used per GSIM region. This resulted in five regions using Viewfinder DEM and 19 regions using HydroSHEDS (see Table 4).

The HydroSHEDS documentation (Lehner et al., 2006) was followed to generate flow direction and flow accumulation datasets, which form inputs for the catchment delineation algorithm along with the specification of a catchment outlet. A python script was developed to automatically call the delineation algorithm in the ArcGIS toolset (Jenson and Domingue, 1988) for each gauge and for a range of options for the outlet location. Two different cases occurred when identifying the 'best location' of the catchment outlet.

**Case 1. Reported station coordinates adopted as the outlet**

If there was no information about a drainage area in the station metadata, the geographical coordinates of the station were used as the outlet of the delineation process. There are automated techniques for repositioning outlets, such as choosing cells with the greatest flow accumulation within a search-distance (Snap Pour Point ArcGIS tool), or finding the nearest cell possessing a flow-accumulation value above a specified threshold (Lindsay et al., 2008). Nonetheless, without additional reported information on the catchment (such as catchment area, or information extracted from the station name) it is difficult to assess the quality of the delineated catchment. Even if a repositioning technique were adopted, delineated catchment boundaries should be used with caution.

**Case 2. Application of an automated repositioning algorithm**

For stations with available information on catchment area, the automated repositioning procedure documented in GRDC report number 41 (Lehner, 2012) was used with some minor adjustments and is summarised below:

(1) The catchment area was estimated using the flow accumulation dataset derived from the DEM products. This calculation was repeated for all pixels of the HydroSHEDS/Viewfinder gridded river network within a search radius of 5 km from the geographical coordinates of a specific station.

(2) The estimated area values were compared with the reported area in the original metadata. All pixels were coded with the absolute value of their area differences (in %, with reported area in the metadata was used as reference). Pixels with area differences of more than 50% were excluded. This procedure provided an area-based ranking scheme ($RA$) ranging between 0 and 50, where 0 indicates perfect agreement in catchment areas.

(3) The distance to the original location of the station (geographical coordinates reported in original metadata) was calculated for each pixel and normalised to reach 50 at the maximum distance of 5 km. This procedure provided a distance-based ranking scheme ($RD$) ranging between 0 and 50, where 0 indicates perfect agreement in station locations.

(4) The final ranking scheme ($R$) was calculated as a combination of $RA$ and $RD$, where distance rank was weighted twice as high ($R = RA + 2RD$) to additionally penalise pixels that were further away from the original location.

(5) The outlet was automatically relocated to the position of the pixel showing the lowest ranking value, and geographical coordinates of the pixel centroid were defined as the 'best' outlet for this specific catchment.

(6) In the original GRDC technical document (Lehner, 2012), a manual procedure was adopted for stations with differences in area above 50 (i.e. the search algorithm cannot find any pixel with an area difference less than 50% within the 5 km search radius), or for stations that had no reported area in the GRDC catalogue. This manual inspection process was infeasible given the scope of the GSIM project, having over 30,000 catchments being delineated and where river names were not available (or potentially inaccurately translated) for many stations.





The delineated catchment boundary for each station was assigned a quality flag according to four groups and using discrepancy in catchment area as the main filtering criteria. These groups were:

(1) 'High' quality: Area difference less than 5%

(2) 'Medium' quality: Area difference from 5% to less than 10%

(3) 'Low' quality: Area difference from 10% to less than 50%

(4) 'Caution' quality: Area difference higher than or equal to 50%, or the reported catchment area was not available in the GSIM catalogue.

**4.3 Extraction of catchment-scale metadata**

An important aspect of large-scale hydrology is the ability to exploit gridded datasets at the global scale (Bierkens, 2015; Bierkens et al., 2015; Gudmundsson and Seneviratne, 2015; Seneviratne et al., 2012; Ward et al., 2015). Having developed

catchment boundaries for each GSIM station enabled a supplementary set of catchment-scale metadata to be derived with relative ease. A key feature is that the catchment boundaries and the subsequent metadata relates to the upstream contributing area that influences a gauge, rather than to the catchment (or arbitrarily defined sub-catchment) that contains the gauge and therefore includes a non-influencing downstream region.

In developing the catchment-scale metadata, a standard set of variables have been identified with a view to supporting a

range of applications such as filtering stations according to characteristic features, performing analyses of streamflow according to explanatory features of a catchment, or classifying stations according to (in)significance of human impact. As summarised in Table 5, a total number of 12 global data products were used to derive 19 elements of catchment-scale metadata. These products were chosen to represent five main categories of catchment characteristics: (1) topography, (2) human impact, (3) climate type, (4) vegetation type, and (5) soil profile. Because the global data products have varying

resolution and structure, the following method was used to derive the catchment-scale metadata:

(1) Delineated catchment boundaries associated with each stream gauge were used to mask the subset of pixels from the resampled dataset.

(2) If more than 30% of the catchment area was not covered by a specific global data product, a 'No data' code was given.

(3) Metadata representing the characteristics of the upstream catchment for each streamflow gauge were calculated from the gridded data masked in step (1). There were three types of metrics calculated during this step:

(a) **A single value**. Used only for the elevation at the geographical coordinates of the gauge (i.e. the catchment outlet), number of large dams located within the catchment boundary, and total volume of corresponding reservoir.

(b) **Average, min, max and quartiles values.** Used for continuously varying data such as a slope or topography index. These metrics allow an idea of central tendency as well as spread of extracted data within each catchment boundary.

(c) **Percentages of different classes of catchment characteristics.** Used for categorical data. For example, there are 16 classes in the global lithology dataset, and the co-presence of more than one type of lithology

occurs very often across all catchments. The percentages of each lithology class were therefore calculated and recorded for all available catchments. To make the results presentable in a final catchment-scale metadata matrix, an aggregated metric was calculated to indicate that there is a dominant class within the catchment boundary (i.e. more than 50% of all available pixels). If there is no dominant class within the catchment boundary, a 'No dominant class' string is provided.



### 5. Overview of the GSIM archive

This section summarises the GSIM archive, including the availability of timeseries combined from 12 original data sources, the associated data-products and documentation outlining data quality.

#### 5.1 Timeseries availability

From the total number of 35,002 timeseries records obtained from 12 different sources, the final GSIM timeseries archive holds a total number of 30,959 unique stations, of which 30,935 stations have associated catchment shapefiles and catchment-scale metadata (24 stations were removed from this process due to suspect geographical locations). As shown in Table 6, it is apparent that spatial coverage of the stations in the GSIM database varies significantly across continents, with North America and Europe having the greatest number of stations.

Including the national databases such as MLIT (Japan), ANA (Brazil), BOM (Australia), and IWRIS (India) has significantly improved the observational network over the regions of Asia, South America and Oceania, some of which have recorded streamflow since the mid-20th century and were still operating at the time the GSIM database was initiated. This suggests that the national databases that are currently available should be given more attention to improve the quality and quantity of international archives.

Regarding temporal coverage, streamflow records across the globe are generally available for the second half of the 20th century (as shown in the bottom panel of Fig. 4). Regardless of missing data criteria, the number of available data gradually rises to its peak in the late 1970s to early 1980s, followed by a mild decrease in the late 1980s (Hannah et al., 2011) and a secondary peak in the late 2000s. While the overall database has over 30,000 gauges, it is clear from Fig. 4 that from the 1960s onwards there are approximately from 10,000 to 15,000 gauges simultaneously active. This represents a significant increase in availability compared to the GRDC dataset, which had a total of approximately 9000 gauges and with a similar drop-off in available gauges depending on the filtering criteria applied.

#### 5.2 Data products of GSIM

A range of data products has been made available as part of GSIM.

#### 5.2.1 GSIM catalogue

The GSIM catalogue (provided as supplementary material to this paper) is designed for users to easily filter stations according to their purpose of application and where necessary, to transparently identify steps taken in the development of GSIM. The total number of 27 fields included in this document can be divided into three groups, namely:

(1) Basic metadata: This group provides station identification, including a unique GSIM number, the name of the river, the name of the station, the elevation of the gauge, the geographical coordinates, and the catchment area.

(2) Database merging metadata: This group of fields provides the identity of the numbers of original source(s) and if applicable, the similarity metrics between duplicates.

(3) Data availability metadata: This group of fields provides an overview on the data availability of each individual timeseries. These statistics were generated from the timeseries data and can be used to filter station information, such as temporal coverage, data length, and the fraction of missing data.

As illustrated in Table 7, source datasets had significant gaps in the metadata, especially in cases of gauge elevation (not available in CHDP, GAME, HYDAT, BOM, MLIT) and catchment area (not available in BOM, MLIT). In addition, the geographical coordinates of all stations were not correctly recorded for all stations, with 24 removed as having suspect locations and 4,871 shifted coordinates as part of the procedure for aligning catchment outlets with reported catchment areas.



### 5.2.2 Quality of catchment boundary

The catchment boundary is the second metadata product that has been made available through GSIM. Of all GSIM stations, 12,150 (39%) were not associated with any information about drainage areas (including all MLIT and BOM stations); thus, a 'Caution' flag is attached to upstream catchments of these stations. Another 24 stations with suspected geographical coordinates of stations were also removed, and the final 18,785 stations were processed to identify 'best outlet' location to represent the outlet for delineating upstream catchments. The distribution and quality of the delineated

catchments of these stations are provided in Fig. 5.

As illustrated in the top panel, 'Caution' catchments using 'best' outlets (identified using the method outlined in Section 4.2) are generally located across all GSIM regions. However, the 'Caution' flag appears more frequently over regions above 60 degrees North. Further checks would be required to improve the association of catchment boundaries to stations. Unfortunately, the biggest caveat that applies to the GSIM database, as with any global database, is that the metadata

were collated from a number of sources with varying standards of documentation and quality assurance and with limited capacity for additional checking other than automated procedures. Therefore, there is likely to be a non-trivial degree of error in the metadata for both geographical location and drainage area. Another issue that may lead to unreliable results of the delineation process is error in the DEM products. This potential error has been documented (Lehner, 2012; Lehner et al., 2006), and lower quality in DEM products generally exist for regions above 60 degrees North due to the lower

quality of the original elevation products used to derive the DEM datasets. Another note for the use of delineated catchments is that very small catchments (area less than 50 km$^2$) should be handled with care, as the 'best' outlets could be located incorrectly while still delivering 'acceptable' discrepancies as part of the automated procedure.

Nonetheless, the quality of delineated catchments is quite positive (as illustrated in Fig. 7). Of all 18,785 catchments that had reported drainage area in the GSIM catalogue, 68.25%, 11.8% and 15.92% catchments have 'high' quality (area

discrepancy of less than 5%), 'medium' quality (area discrepancy from 5% to less than 10%) and 'low' quality (area discrepancy from 10 to less than 50%) respectively, while there are only 4.03% catchments with 'Caution' quality (area discrepancy of more than or equal to 50%).

### 5.2.3 Catchment-scale characteristics availability

The final data product that has been made available is the auxiliary information extracted from 12 global coverage datasets

representing many characteristics associated with GSIM stations. Overall, the spatial coverage of original data products (mostly satellite-base) are quite good (see Table 8), with just a small fraction of catchments (less than 10%) that have more than 30% of their areas not covered by these datasets. The exception is the Nightlight Development Index (NLDI – computed from the 2006 Nightlights dataset (Ziskin et al., 2010) and the year 2006 Landscan gridded population (Bhaduri et al., 2002)). This dataset does not have approximately 25.3% of catchments covered, for more than 70% of their areas.

It is important to note that while these catchment-scale characteristics are consistent products available for all stations, documentation for the original source data should be consulted during application to appreciate the limitations of each variable. For example, the GRanD database is not exhaustive of all dams worldwide and there can be ambiguities over the affiliated dates (e.g. whether they represent conception, construction or commissioning). Similarly, it is likely that there will be updated or new data gridded datasets available over time so that applications should consider the

appropriateness of the information used. The availability of metadata products emerging from the GSIM project demonstrates the possibility of using reported global data products to extract catchment-scale characteristics associated with each station with reasonable quality, enabling many potential applications from this rich information.



**6 Data availability**

The data described in this paper are available as a compressed zip-archive containing (i) a readme file, (ii) metadata of all GSIM stations obtained from original data sources and time series, (iii) quality of catchment boundary and catchment characteristics extracted from 12 global data-products and (iv) catchment boundaries for 30,935 stations that have reasonable geographical location.

The data can be freely downloaded at https://iacweb.ethz.ch/staff/lukasgu/GSIM/GSIM_metadata.zip during the peer-reviewing phase. Conditional on acceptance of this article, the data will be made available on the www.pangea.de data

server and will be accessible through a digital object identifier (doi).

**7 Conclusion**

*In-situ* observations of daily streamflow with global coverage are crucial to understanding large-scale freshwater resources that are fundamental for societal development. The GSIM archive, designed as an expansion of the GRDC database, has demonstrated the possibility of significantly improving the coverage and density of the global streamflow

observational datasets using free-to-access databases. The development of the GSIM database was not possible without the tremendous investment into the production and ongoing maintenance of original data sources of GSIM. This fact emphasises the key role of data authorities and international initiatives in enabling advances in large-scale hydrology by making data publicly available to the community.

While the activities of GSIM have been extensive in searching out and collating databases, they are by no-means

exhaustive (e.g. since submission we have been notified of additional potential candidates for inclusion such as the Mekong River Commission database, Chile national water database, Argentina national water database). It is the authors' intention that this project will stimulate further efforts toward the development of coordinated and consistent representation of global streamflow observations. For this reason, the process of developing the archive was designed with automation in mind. With the exception of needing to visually inspect some cases of duplicated timeseries, the

archive was automated using scripts in the R and Python programming languages.

Although the GSIM database was compiled from data sources that can be obtained free of charge via a data portal or by submitting written requests to data authorities, there are some strict conditions related to the redistribution of un-processed data. Therefore, it is impossible to make the whole GSIM collection publicly available. In addition, with the main aim of harvesting as much data as possible, the GSIM database is not focused on collecting high-quality datasets such as

referenced hydrological networks that are available in many countries (Whitfield et al., 2012), and thus the data quality may vary significantly across the available timeseries. To address these limitations and increase the usefulness of the GSIM database, we conducted a set of quality checking procedures for all GSIM timeseries. These quality-assured records were then used to produce a dedicated set of indices capturing important aspects of the daily dynamics from GSIM timeseries, and to explore potential applications of GSIM in large-scale hydrology. Detailed information about this work

and associated distributed data is described in the second part of our series on GSIM (Gudmundsson et al., submitted).

With the GSIM archive and production information made publicly available in a transparent manner, this project serves the broader hydrology community with improved coverage and quality of streamflow information. This project has yielded a significant increase in the availability of streamflow observations through the process of collating readily-accessed online data, and with ongoing efforts there will be opportunities for further extension. Streamflow observations

represent an underutilized resource, in part due to access limitations, but also due to challenges in accounting for human impacts in the observed record. These challenges notwithstanding, ongoing advances in global-scale hydrological models



and ever-increasing access to remote-sensed products indicate that wider access to streamflow data has the potential to significantly enhance our knowledge of global water resources.






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

120



**Tables**

**Table 1. Basic information of daily streamflow databases included in BSDB project**

| Database (referred name) | Database category | Spatial coverage | Data access information |
|---|---|---|---|
| Global Runoff Data Centre (GRDC) | Research database | Global | www.bafg.de/GRDC/ Archived database can be obtained via written request to GRDC. |
| European Flow Regimes from International Experimental and Network Data (EWA) | Research database | European | http://ne-friend.bafg.de/servlet/is/7413/ Data can be ordered for EURO-FRIEND-Water project purposes free of charge. |
| A Regional, Electronic, Hydrographic Data Network for Russia (ARCTICNET) | Research database | Russia | http://www.russia-arcticnet.sr.unh.edu/ Archived database |
| China Hydrology Database Project (CHDP) | Research database | China | http://www.oberlin.edu/faculty/aschmidt Archived database can be obtained via written request to the author of the database |
| GEOSS ana MAHASRI Experiment in Tropics (GAME) | Research database | Thailand | http://hydro.iis.u-tokyo.ac.jp/GAME-T/GAIN-T/routine/rid-river/disc_d.html Archived database |
| U.S. National Water Information System (USGS) | National database | USA | http://waterdata.usgs.gov/nwis Individual timeseries can be downloaded from the data portal |
| Canada National Water Data Archive (HYDAT) | National database | Canada | https://ec.gc.ca/rhc-wsc/ Archived database |
| Brazil National Water Agency (ANA) | National database | Brazil | http://hidroweb.ana.gov.br/ Individual timeseries can be downloaded from the data portal |
| Japan Water Information System (MLIT) | National database | Japan | http://www1.river.go.jp/ Individual timeseries can be downloaded from the data portal |
| Anuario de aforos digital 2010 - 2011 (AFD) | National database | Spain | http://ceh-flumen64.cedex.es/anuarioaforos Archived database, DVD available from Spanish authorities |
| Australia Water Data Online (BOM) | National database | Australia | http://www.bom.gov.au/waterdata/ Individual timeseries can be downloaded from the data portal |
| Water Resources Information System of India (I-WRIS) | National database | India | http://www.india-wris.nrsc.gov.in/wris.html Individual timeseries can be downloaded from the data portal |

125



**Table 2. Number of stations in countries where national databases are available**

| Country | Database | | |
|---|---|---|---|
| | EWA | GRDC | National |
| Australia | - | 358 | 2941 (BOM) |
| Brazil | - | 439 | 3313 (ANA) |
| Canada | - | 1029 | 6325 (HYDAT) |
| India | - | 0 | 318 (WRIS) |
| Japan | - | 151 | 1029 (MLIT) |
| Spain | 239 | 0 | 1197 (ADF) |
| United State | - | 981 | 9404 (USGS) |





**Table 3. Basic metadata available from data sources**

| Database | Station ID | Station name | River name | Geographical coordinates | Station elevation | Drainage area | Catchment boundary |
|----------|-----------|-------------|-----------|------------------------|------------------|--------------|-------------------|
| GRDC | X | X | X | X | X | X | X |
| EWA | X | X | X | X | X | X | X |
| CHDP | X | X | X | X | - | X | - |
| GAME | X | X | X | X | X | X | - |
| ARCTIC NET | X | X | X | X | X | X | - |
| USGS | X | X | - | X | X | X | - |
| HYDAT | X | X | - | X | - | X | - |
| ANA | X | E | E | X | X | X | - |
| ADF | X | E | E | X | X | X | - |
| MLIT | X | E | E | X | - | - | - |
| BOM | X | X | - | X | - | - | - |
| WRIS | X | X | X | X | X | X | - |

*(x: metadata available; -: metadata is unavailable; e: metadata is not available in English)*



**Table 4. DEM product used for each GSIM region**

| Region | Description | DEM product |
|---|---|---|
| Artic (region 1) | Represents the distant part of North America (including Alaska, most part of Canada and eastern part of Autonomous Province, Russia) | Viewfinder DEM 15s |
| Europe above 60N (region 2) | Represents countries located above 60°N (e.g. Sweden, Denmark, Norway, part of Germany, part of Russia) | Viewfinder DEM 15s |
| Siberia (region 3) | Represents areas above the 60°N part of Asia | Viewfinder DEM 15s |
| Islands (region 4) | Represents some islands across Pacific Ocean (e.g. Honolulu, U.S.) and Atlantic Ocean | Viewfinder DEM 15s |
| Greenland (region 5) | Represents land mass of Greenland | Viewfinder DEM 15s |
| Europe 1 to Europe 6 (six regions, from region 6 to region 11) | Represent most European countries (below 60°N) | HydroSHEDS DEM 15s |
| North America 1 to North America 9 (nine regions, from region 12 to region 20) | Represent U.S. (except Alaska) and the southern part of Canada (below 60°N). It also includes Central America for simplicity in processing catchment boundaries. | HydroSHEDS DEM 15s |
| Africa (region 21) | Represents Africa region | HydroSHEDS DEM 15s |
| Asia (region 22) | Represents Asia region (part of Kazakhstan, China, Mongolia and Russia) | HydroSHEDS DEM 15s |
| Australia (region 23 | Represents Australia, New Zealand and some pacific islands | HydroSHEDS DEM 15s |
| South America (region 24) | Represents South America | HydroSHEDS DEM 15s |



**Table 5. Global data-products used in GSIM and derived catchment-scale metadata**

| Variables | Data sources | Resolution | Extracted metadata |
|---|---|---|---|
| Elevation | HydroSHEDS http://hydrosheds.org/ ViewFinder http://viewfinderpanoramas.org/ | 15 arc-seconds x 15 arc-seconds | (1) Gauge elevation (2a-f) Average, minimum, maximum, first quartile, second quartile and third quartile values of catchment elevation |
| Slope | Derived from HydroSHEDS and ViewFinder D.E.M by authors | 15 arc-seconds x 15 arc-seconds | (3a-f) Average, minimum, maximum, first quartile, second quartile and third quartile values of catchment slope |
| Topographic index | High-resolution global topographic index values (Marthews et al., 2015) https://catalogue.ceh.ac.uk/documents/ce391488-1b3c-4f82-9289-4beb8b8aa7da | 15 arc-seconds x 15 arc-seconds | (4a-f) Average, minimum, maximum, first quartile, second quartile and third quartile values of catchment topographic index |
| Drainage density | GRIN - Global River Network (Schneider et al., 2017) https://www.metis.upmc.fr/fr/node/375 | 7.5 arc-minutes x 7.5 arc-minutes | (5a-f) Average, minimum, maximum, first quartile, second quartile and third quartile values of catchment drainage density ($km^{-1}$) |
| Dams | Global Reservoir and Dam (GRanD), version 1 (Lehner et al., 2011) http://sedac.ciesin.columbia.edu/data/set/grand-v1-dams-rev01 | 6,862 datapoints storage capacity of more than 0.1 km3 | (6) Number of dams upstream (7) Total upstream storage volume |
| Population | Gridded Population of the World (GPW) version 4 (CIESIN, 2016) http://sedac.ciesin.columbia.edu/data/set/gpw-v4-population-count | 30 arc-seconds x 30 arc-seconds | (8a-f) Average, minimum, maximum, first quartile, second quartile and third quartile values of catchment population (2010) (9) 2010 Population count |
| Urbanisation | Night Light Development Index (NLDI) dataset (Elvidge et al., 2012) http://www.soc-geogr.net/7/23/2012/sg-7-23-2012.html | 0.25 arc-degrees x 0.25 arc-degrees | (10a-f) Average, minimum, maximum, first quartile, second quartile and third quartile values of NLDI over catchment |
| Irrigation | Historical Irrigation Dataset (Siebert et al., 2015) https://mygeohub.org/publications/8/2 | 5 arc-minutes x 5 arc-minutes | (11a-f) Average, minimum, maximum, first quartile, second quartile and third quartile values of catchment Irrigated area (2005) |
| Climate type | World map of Koppen Weiger climate classification system (Rubel and Kottek, 2010) http://koeppen-geiger.vu-wien.ac.at | 5 arc-minutes x 5 arc-minutes | (12) Type of catchment climate (Koppen-Weiger) if one type present over more than 50% catchment area, or 'No dominant type' |
| Land cover | The Climate Change Initiative Land Cover (CCI-LC) dataset http://maps.elie.ucl.ac.be/CCI/viewer/download.php | 7.5 arc-seconds x 7.5 arc-seconds | (13) Type of catchment land-cover (UN Land Cover Classification System) for 2015 if one type present over more than 50% catchment area, or 'No dominant type' |



| Variables | Data sources | Resolution | Extracted metadata |
|---|---|---|---|
| Lithological | The Global Lithological Map v1.0 (GLiM) dataset (Hartmann and Moosdorf, 2012) https://www.clisap.de/research/b:-climate-manifestations-and-impacts/crg-chemistry-of-natural-aqueous-solutions/global-lithological-map/ | 0.5 arc-degrees x 0.5 arc-degrees | (14) Type of catchment lithology if one type present over more than 50% catchment area or 'No dominant type' |
| Soil profile | Soil grid 250m (Hengl et al., 2017) https://soilgrids.org | 7.5 arc-seconds x 7.5 arc-seconds | (15) Type of catchment soil-class (World Reference Base) if one type present over more than 50% catchment area or multiple types 'No dominant type'. (16a-f) Average, minimum, maximum, first quartile, second quartile and third quartile values of weight percentage of sand over the catchment (17a-f) Average, minimum, maximum, first quartile, second quartile and third quartile values of weight percentage of silt over the catchment (18a-f) Average, minimum, maximum, first quartile, second quartile and third quartile values of weight percentage of clay over the catchment (19a-f) Average, minimum, maximum, first quartile, second quartile and third quartile values of bulk content of soil over the catchment ($kg/m^3$) |





**Table 6. Summary statistics of GSIM timeseries**

| Continent | Number of stations | Average temporal coverage (years) | Shortest record (years) | Longest record (years) | Year of earliest entry | Year of latest entry |
|---|---|---|---|---|---|---|
| Africa | 949 | 33.8 | 1 | 110 | 1903 | 2015 |
| Europe | 5,778 | 40.3 | 1 | 208 | 1806 | 2016 |
| Asia | 1,915 | 22.2 | 1 | 79 | 1921 | 2015 |
| North America | 15,884 | 42.9 | 1 | 156 | 1860 | 2016 |
| South America | 3,449 | 29.3 | 1 | 116 | 1901 | 2016 |
| Australia and Oceania | 2,984 | 31.4 | 1 | 131 | 1886 | 2016 |
| **Global** | **30,959** | **38.2** | **1** | **208** | **1806** | **2016** |



**Table 7. The percentage of stations accompanied by all basic metadata**

| Dataset | Station ID | River name | Station name | Latitude | Longitude | Altitude | Catchment area |
|---|---|---|---|---|---|---|---|
| ADF | 100 | 100 | 100 | 100 | 100 | 96.2 | 99.3 |
| ANA | 100 | 99.9 | 100 | 100 | 100 | 69 | 99 |
| ARCTICNET | 100 | 100 | 100 | 99.3 | 99.3 | 99.3 | 100 |
| BOM | 100 | 100 | 100 | 100 | 100 | 0 | 0 |
| CHDP | 100 | 99.4 | 100 | 100 | 100 | 0 | 84 |
| EWA | 100 | 100 | 100 | 100 | 100 | 98.5 | 94.5 |
| GAME | 100 | 100 | 100 | 100 | 100 | 0 | 100 |
| GRDC | 100 | 100 | 100 | 100 | 100 | 67 | 100 |
| HYDAT | 100 | 100 | 100 | 100 | 100 | 0 | 85.8 |
| MLIT | 100 | 100 | 100 | 100 | 100 | 0 | 0 |
| USGS | 100 | 100 | 100 | 100 | 100 | 93.7 | 25.5 |
| WRIS | 100 | 100 | 100 | 100 | 100 | 81.6 | 97.4 |
| **GSIM** | **100** | **99.9** | **100** | **99.9** | **99.9** | **50.4** | **73.8** |



**Table 8. Percentages of available catchment-scale characteristics**

| Catchment characteristics | Number of stations | Availability percentage |
|---|---|---|
| Climate classification | 30,773 | 99.5 |
| Drainage density | 29,574 | 95.6 |
| Elevation | 30,932 | 99.9 |
| Irrigation area | 30,857 | 99.7 |
| Land cover classification | 30,888 | 99.8 |
| Lithology type | 30,154 | 97.5 |
| Nightlight Development Index | 23,096 | 74.7 |
| Population count | 30,894 | 99.9 |
| Population density | 30,800 | 99.6 |
| Slope | 30,862 | 99.8 |
| Soil bulk density | 30,812 | 99.6 |
| Soil classification | 30,764 | 99.4 |
| Clay content | 30,768 | 99.5 |
| Clay content | 30,695 | 99.2 |
| Silt content | 30,828 | 99.7 |
| Topographic index | 30,725 | 99.3 |





**Figures**

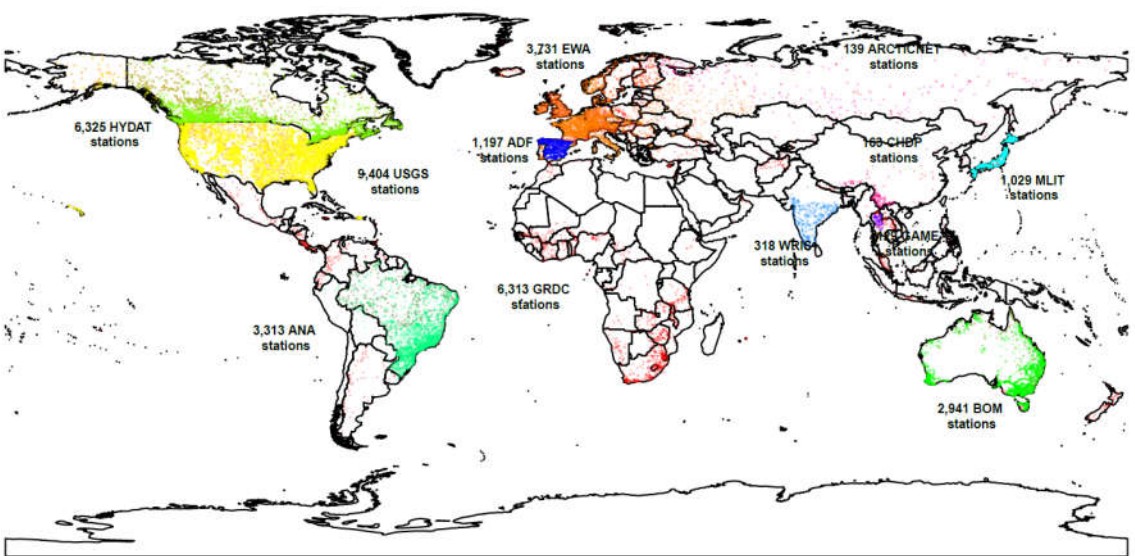

**Figure 1. The distribution of stations from original data sources**




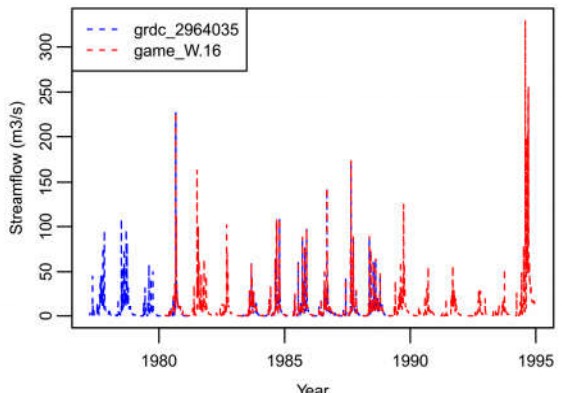
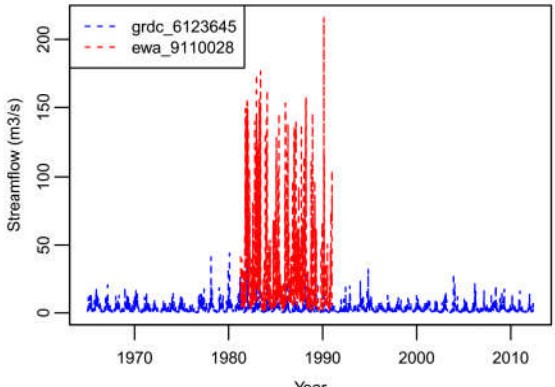

**Figure 2. Examples of visually inspected duplication-candidate timeseries. Left: Two stations were labelled 'very likely identical' stations. Right: Two stations were labelled 'very likely different' stations.**





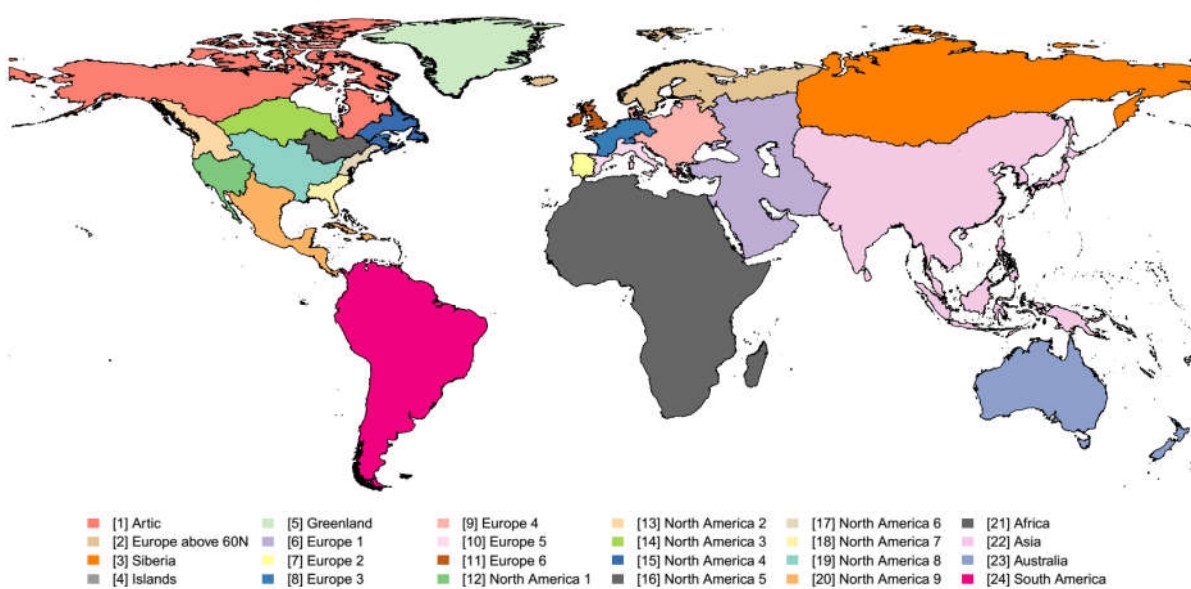

**Figure 3. GSIM regions for catchment delineation and metadata extraction procedures**





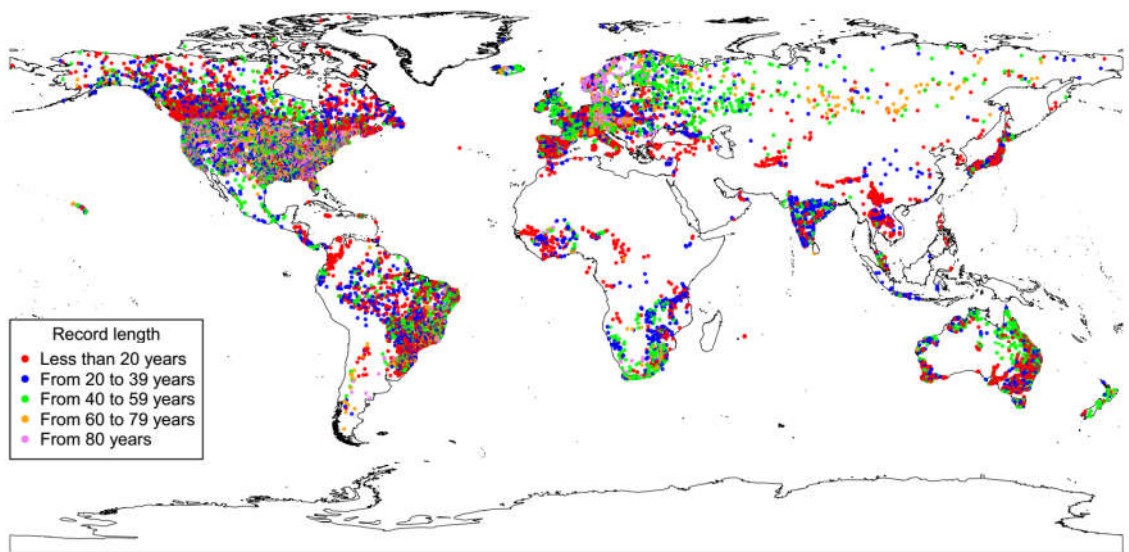

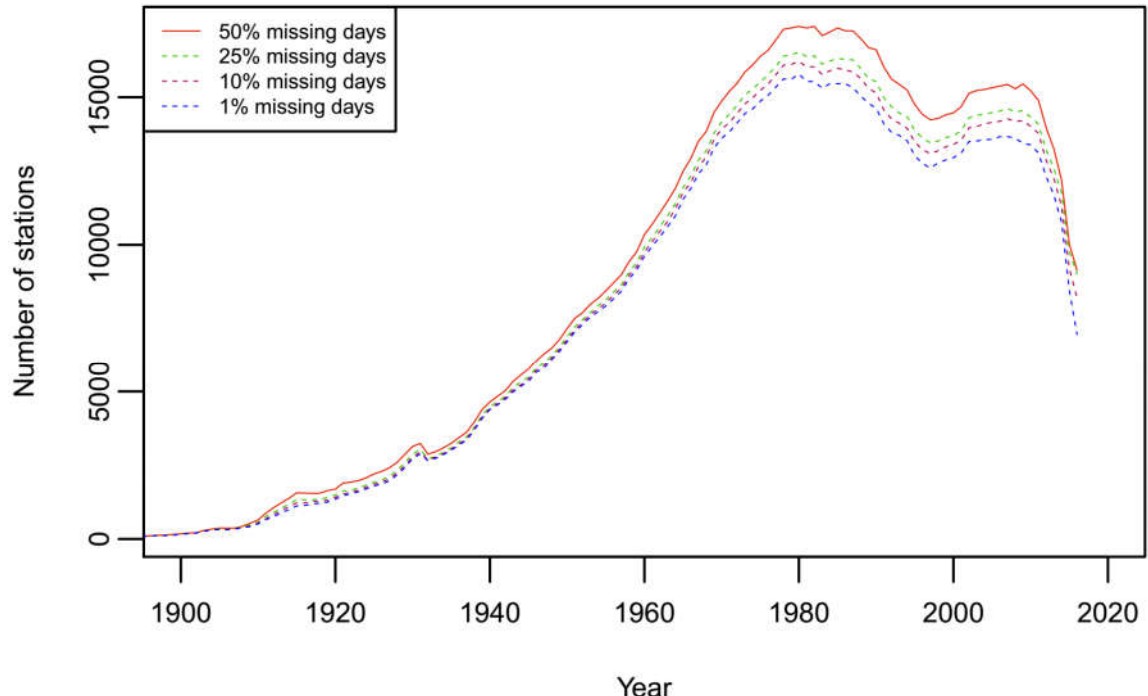

**Figure 4. Availability of GSIM timeseries. The top panel illustrates the length of record at each station, and the bottom panel illustrates the available time series over time for four different missing data criteria.**



**Figure 5. Quality of delineated catchment boundary according to categories of high, medium low and caution identified in Section 4.2 (for 18,785 stations that have reported drainage area and reasonable geographical coordinates).**