# Peer review of "The Global Streamflow Indices and Metadata Archive (GSIM) – Part 1: The production of daily streamflow archive and metadata"

_Earth System Science Data, 2017_

## Short Comment (SC1) · 13 Dec 2017

Great data set! Will be very useful. Would be great if the underlying daily time-series data can also be made available via an online repository?

Looking at the contents of file GSIM_metadata.csv, I think that the units for the attribute "altitude" are in *feet* for USGS gauges? E.g. gauge US_0006679 has an altitude value of 10520. Would be better to correct this to meters, so it is consistent with the altitude of gauges from other sources (e.g. GRDC).

I have not checked the attribute "area" of USGS gauges. Please ensure it is in $km^2$.

[Figure]

I also noticed that at least one gauge has impossible lat/lon coordinates: KG_0000003 (lat/lon: 90/180) If location is unknown would be better to assign NA value. I have not checked if the same applies to many other gauges with unknown location.

regards, Harald Kling

---

## Short Comment (SC2) · 13 Dec 2017

In file GSIM_metadata.csv the ID number for all Japanese gauges is always "3.04E+14". Probably a formatting issue, which should be corrected.

I came across this by chance. I suggest that the authors thoroughly check all their attributes before final publication of the data.

---

## Referee Comment (RC1) · W. Grabs (Referee) · 18 Dec 2017

ESSD- 2017- 103

Review

Wolfgang E. Grabs

20 Insert references on global runoff estimation using global discharge data sets

35 GRDC operating under the auspices of the UN - World Meteorological Organization (WMO)

[Figure]

40 Update statistics from current GRDC catalogue

95 ARCTICNET is a now static database that is mirrored in the GRDC. In GRDC, stations of ARCTICNET are updated based on data deliveries to GRDC. The ARCTIC-HYCOS river discharge network is hosted and operated by GRDC with currently over 500 stations that are online available.

115 Most data of the European Water Archive (EWA) hosted by GRDC are available under the GRDC data policy and are no longer restricted to the FRIEND data policy. Based on data deliveries, the EWA is updated.

140 ARCTICNET is superseded by the arctic river basin database and ARCTIC-HYCOS databases that are part of the GRDC database. However, the ARCTIC-HYCOS database is operated as a project and data are open.

230 It needs to be noted that data deliveries from national official data suppliers also contain errors. GRDC is performing plausibility checks on these data sets to detect and correct errors and provide feed-back to data suppliers.

Question: Have there been some checks to detect consistency of data sets supplied by national suppliers with data sets from the same stations contained in the GRDC?

It is always necessary to check for the latest available versions of databases!

280 As standardization issues are a prominent issue it its important to describe if the development of the metadata catalog has followed standrds set (and endorsed by WMO as a standard setting organization), by the Open Geospatial Consortium (OGC), using WATER ML-2

320 Is GSIM Metadata compliant with OGC standards?

435 ff Provide information on the status of time series. In the case of archived time-series: describe whether there are update mechanisms in place or whether some of the data are closed data sets. There is the danger to generate orphaned data sets

with incomplete information (metadata) on the version and last date of such data sets. This has often created confusion as researchers worked with outdated data sets (such as the UNESCO RivDis that still is used although it is outdated since over 20 years. It is used as the data is open without restrictions but the data holdings contain errors and/or have long since been replaced or updated including error correction.

510 Explain in a more transparent manner that GSIM will provide the metadata archive and not the actual time series as a result of different data policies from database operators including national services.

Discuss in more detail existing update mechanisms of databases and an indication which data sets are closed historic archives and which are living databases that are continuously updated.

Corrections in tables:

Table 1 ARCTICNET is part of GRDC, in addition, GRDC hosts the ARCTIC-HYCOS database; ARCTICNET is a closed historic database

Table 2 Example for Spain: EWA has 239 stations, GRDC 0 BUT: EWA is hosted under GRDC and data are available under GRDC data policy. These are no longer separated data bases!

20, citation of global discharge:

http://www.bafg.de/GRDC/EN/03_dtprdcts/33_CmpR/unh_grdc_node.htmlPlease cite in your publication the GRDC as the source of the data: Fekete, B., Vörösmarty, C., and W. Grabs (2002): Global composite runoff fields on observed river discharge and simulated water balances / Water System Analysis Group, University of New Hampshire, andGlobal Runoff Data Centre. Koblenz, Germany : ...

High‐resolution fields of global runoff combining observed river ... http://onlinelibrary.wiley.com/doi/10.1029/1999GB001254/pdfHigh-resolution fields of global runoff combining observed river discharge and simulated water balances.

Balázs M. Fekete and Charles J. Vörösmarty1. Complex Systems Research Center, Institute for the Study of Earth, Oceans, and Space, University of New Hampshire,. Durham, New Hampshire, USA. Wolfgang Grabs.

Charles J. Vörösmarty - Google Scholar Citations http://scholar.google.com/citations?user=5kSVDe4AAAAJ&hl=deHigh‐resolution fields of global runoff combining observed river discharge and simulated water balances. BM Fekete, CJ Vörösmarty, WGrabs. Global Biogeochemical Cycles 16 (3), 2002. 411, 2002. A continental–scale model of water balance and fluvial transport: Application to south America. CJ Vörösmarty, B MooRe, ...

Welcome To UNH/GRDC Composite Runoff Fields V1.0 http://www.grdc.sr.unh.edu/Balázs M. Fekete Charles J. Vörösmarty Wolfgang Grabs. Report and Supporting Documentation Âů Station Data Explorer Âů Basin Data Explorer Âů Runoff Data Explorer Âů Data Download (Monitoring Stations, River Networks & Runoff Fields). Database Supporters, Co-sponsors. Return to Water Systems Research Group Home ... Website, downloads etc...)

280 Metadata data exchange standard

OGC® WaterML | OGC http://www.opengeospatial.org/standards/waterml1) Overview Âů 2) Documents and Downloads Âů 3) Official Schemas Âů 4) Related News. 1) Overview. WaterML 2.0 is a standard information model for the representation of water observations data, with the intent of allowing the exchange of such data sets across information systems. Through the use of existing OGC standards, ...

The 5 Essential Elements of a Hydrological Monitoring Programme ... https://public.wmo.int/en/bulletin/5-essential-elements-hydrological-monitoring-programmeFor example, the Water ML2.0 standard provides for the exchange of (1) point- based time series data, (2) processed values such as forecasts and ... and stored out of harm's way, (2) metadata are complete, and (3) documentation is available for any changes in methods that could potentially impact the integrity of the

data.

---

## Referee Comment (RC2) · Anonymous Referee #2 · 31 Dec 2017

The paper describes the harmonised station metadata and catchment characteristics of a merged global river discharge dataset. There will be different opinions regarding the approach 'more will offset potential quality deficits' versus a 'less is more' strategy that is often applied (and necessary) for specific research. This could perhaps still be discussed a little bit better in this paper's intro. Nevertheless, I found the material well presented and the data will be useful. The steps towards the collation, selection and derivation and processing of the metadata for this large archive are well described. This documentation may help the appreciation of the often invisible but always tremendous effort that goes into harmonized datasets and I would like to highlight in particular the careful consideration and provision of quality flags for the derived metadata in this case.

[Figure]

I hope that this information will be used, rather than overlooked. Perhaps a sentence on this important data aspect could be placed more prominently in the abstract and conclusion. Hopefully, the paper will provide incentive for some national databases, to provide access to the metadata they often have but don't provide as readily, such as catchment boundaries, topographical features and land cover.

A few minor issues that I recommend be addressed are listed below.

line 33 "questions over its utility" - it's not really clear what is meant. If the indended use is climate sensitivity analysis, yes, but there are quite a few other uses. Maybe clarify utility for... or phrase more generally.

line 324ff This section contains a bit of redundant information and a few typos (suggest to proofread again)

For the reader to get an impression of the precision of catchment area delineation, I think it is important to show an zoomed example of some kind.

In 5.2 or in the conclusion I think a bit more discussion or cautionary words should be spent over the fact that there will be catchments in the database for which streamflow time series do not overlap barely or not at all with the time covered by (the relatively new or short) remote sensing based datasets. This requires users to carefully check time overlap for possible cause-effect studies. And ideally metadata readme or column headers should provide the time period covered by the underlying datasets.

Figures 1 and 4 (upper) and 5 (upper) are entirely useless at the resolution and in the jpg format provided in the pdf-download. Dots are indistinguishable. High resolution will be necessary, but likely still not sufficient to make this a useful map. I suggest to create zooms into subdivided regions that will allow to see some of the differences within regions/countries.

Fig 5 lower. Make proper superscripts in the axes labels and change tick labels units e.g. to million or so (or at least also use proper superscripting) - see Journal's

Manuscript guidelines.

---

## Author Comment (AC1) · 12 Jan 2018

We thank the colleagues who have taken their time to provide us suggestions to improve the quality of the metadata products. The full-text of the comments have been included below as *italic text*, followed by our response as normal, indented text.

*Great data set! Will be very useful. Would be great if the underlying daily time-series data can also be made available via an online repository? Looking at the contents of file GSIM_metadata.csv, I think that the units for the attribute "altitude" are in \*feet\* for USGS gauges? E.g. gauge US_0006679 has an altitude value of 10520. Would be better to correct this to meters, so it is consistent with the altitude of gauges from other*

[Figure]

*sources (e.g. GRDC). I have not checked the attribute "area" of USGS gauges. Please ensure it is in $km^2$.*

Thank you very much for picking out this bug in the metadata (e.g. the altitude of US stations), which have been fixed in the revision. Regarding to the attribute "area" of USGS gauges, the unit of this attribute has been converted into $km^2$ as part of the catchment delineation process (for comparison purpose).

*I also noticed that at least one gauge has impossible lat/lon coordinates: KG_0000003 (lat/lon: 90/180) If location is unknown would be better to assign NA value. I have not checked if the same applies to many other gauges with unknown location.*

As mentioned in the manuscript, there are 24 "suspect-stations" with unreasonable geographical coordinates (including KG_0000003), which have been removed from the catchment delineation process (see section 5.1). However, we still keep the coordinates in the metadata to reflect original information provided by data-providers.

Nevertheless, we also see that this may made some confusion to GSIM users, and thus have provided a list of all stations with "suspect geographical coordinates", including station KG_0000003, as an additional file in the published data. This information has also been included into the readme file.

*In file GSIM_metadata.csv the ID number for all Japanese gauges is always "3.04E+14". Probably a formatting issue, which should be corrected. I came across this by chance. I suggest that the authors thoroughly check all their attributes before final publication of the data.*

Thank you for informing us about this bug in the metadata, which has been fixed in the revision. We have thoroughly check all attributes in the latest version of the metadata.

---

## Author Comment (AC2) · 12 Jan 2018

We thank the reviewer for taking their time to provide us constructive comments, which have been included below as *italic text*, followed by our response as normal, indented text.

*The paper describes the harmonised station metadata and catchment characteristics of a merged global river discharge dataset. There will be different opinions regarding the approach 'more will offset potential quality deficits' versus a 'less is more' strategy that is often applied (and necessary) for specific research. This could perhaps still be discussed a little bit better in this paper's intro. Nevertheless, I found the material well*

*presented and the data will be useful. The steps towards the collation, selection and derivation and processing of the metadata for this large archive are well described. This documentation may help the appreciation of the often invisible but always tremendous effort that goes into harmonized datasets and I would like to highlight in particular the careful consideration and provision of quality flags for the derived metadata in this case. I hope that this information will be used, rather than overlooked. Perhaps a sentence on this important data aspect could be placed more prominently in the abstract and conclusion. Hopefully, the paper will provide incentive for some national databases, to provide access to the metadata they often have but don't provide as readily, such as catchment boundaries, topographical features and land cover.*

> We thank the reviewer for suggestions to further improve the manuscript's quality. We have revised our paper (particularly in the abstract, introduction and conclusion) to highlight more prominently (1) the two different approaches in harmonising international databases (e.g. harvesting as much data as possible, which is used in GSIM, versus collating reference hydrology databases, which have been used in some recent publications); and (2) the quality of extracted metadata should be considered when using GSIM.

*A few minor issues that I recommend be addressed are listed below.*

*line 33 "questions over its utility" - it's not really clear what is meant. If the indended use is climate sensitivity analysis, yes, but there are quite a few other uses. Maybe clarify utility for... or phrase more generally.*

> Thank you for your recommendation, we have revised our manuscript to clarify this sentence.

*line 324ff This section contains a bit of redundant information and a few typos (suggest to proofread again). For the reader to get an impression of the precision of catchment area delineation, I think it is important to show an zoomed example of some kind.*

Thank you for your suggestion. We have revised section 4.2 (Catchment delineation procedure) to improve the readability of this particular section. We also add an additional figure to illustrate that using outlet-relocating algorithm has delineated more reasonable catchment boundary (see Figure 1).

*In 5.2 or in the conclusion I think a bit more discussion or cautionary words should be spent over the fact that there will be catchments in the database for which streamflow time series do not overlap barely or not at all with the time covered by (the relatively new or short) remote sensing based datasets. This requires users to carefully check time overlap for possible cause-effect studies. And ideally metadata readme or column headers should provide the time period covered by the underlying datasets.*

We have adjusted the manuscript to ensure users are aware of this limitation of the catchment-scale metadata. We also add a new column ("Reference period") in Table 5 to indicate which dataset has a reference period.

*Figures 1 and 4 (upper) and 5 (upper) are entirely useless at the resolution and in the jpg format provided in the pdf-download. Dots are indistinguishable. High resolution will be necessary, but likely still not sufficient to make this a useful map. I suggest to create zooms into subdivided regions that will allow to see some of the differences within regions/countries.*

We thank the reviewer for the comments to improve the manuscript quality. We agree that figures at regional scale will serve GSIM users better, and have included additional figures as supplementary materials (see attached supplementary document of this interactive comment) and have mentioned this in the manuscript. We, however, would prefer to keep current figures in the manuscript (with higher resolution) to provide an overview on data availability.

*Fig 5 lower. Make proper superscripts in the axes labels and change tick labels units e.g. to million or so (or at least also use proper superscripting) - see Journal's Manuscript guidelines.*

We have revised Figure 5 to match the Journal's standard. We also revised figure 2 to fix similar issue (to change axes label texts from m3/s to m3s-1)

Please also note the supplement to this comment:
https://www.earth-syst-sci-data-discuss.net/essd-2017-103/essd-2017-103-AC2-supplement.pdf

[Figure]

[Figure]

**Figure 1.** Example of catchment delineated using geographical coordinates provided in original metadata and re-located geographical coordinates (for station AR_0000007). As can be seen, the catchment boundary delineated using original coordinates is significantly smaller than that delineated from the re-located coordinates.

**Fig. 1.**

**Supplement:**

[Figure]

**Figure 1.** Overview of GSIM data product over Asia. Top panel: data sources for GSIM; middle panel: length of time series in GSIM database; lower panel: quality of catchment based on discrepancy between the reported catchment area and the delineated catchment area (only stations with reported catchment area have been shown).

[Figure]

**Figure 2.** Overview of GSIM data product over Europe. Top panel: data sources for GSIM; middle panel: length of time series in GSIM database; lower panel: quality of catchment based on discrepancy between the reported catchment area and the delineated catchment area (only stations with reported catchment area have been shown).

[Figure]

**Figure 3.** Overview of GSIM data product over North America. Top panel: data sources for GSIM; middle panel: length of time series in GSIM database; lower panel: quality of catchment based on discrepancy between the reported catchment area and the delineated catchment area (only stations with reported catchment area have been shown).

[Figure]

**Figure 4.** Overview of GSIM data product over South America. Top panel: data sources for GSIM; middle panel: length of time series in GSIM database; lower panel: quality of catchment based on discrepancy between the reported catchment area and the delineated catchment area (only stations with reported catchment area have been shown).

[Figure]

**Figure 5.** Overview of GSIM data product over Australia. Top panel: data sources for GSIM; lower panel: length of time series in GSIM database. As catchment area is not provided in the metadata of original data source, the figure for quality of delineated catchment was not shown here (and thus all catchments delineated in Australia are classified as "Caution").

---

## Author Comment (AC3) · 12 Jan 2018

We thank the reviewer for taking their time to provide us constructive comments, which have been included below as *italic text*, followed by our response as normal, indented text.

**Specific comments**

*20 Insert references on global runoff estimation using global discharge data sets*

Thank you for your recommendation, we have updated the manuscript to include the important application of discharge datasets with three additional

references [1],[2],[3].

[1] Fekete, B., Vörösmarty, C., and Grabs, W.: Global Composite Runoff Fields on Observed River Discharge and Simulated Water Balances/Water System Analysis Group. University of New Hampshire, and Global Runoff Data Centre. Koblenz, Federal Institute of Hydrology (BfG), Koblenz, Germany, Federal Institute of Hydrology (BfG), 2002a. 2002a.

[2] Fekete, B. M., Vörösmarty, C. J., and Grabs, W.: High-resolution fields of global runoff combining observed river discharge and simulated water balances, Global Biogeochemical Cycles, 16, 15-11-15-10, 2002b.

[3] Vörösmarty, C. J., Moore, B., Grace, A. L., Gildea, M. P., Melillo, J. M., Peterson, B. J., Rastetter, E. B., and Steudler, P. A.: Continental scale models of water balance and fluvial transport: an application to South America, Global biogeochemical cycles, 3, 241-265, 1989.

*35 GRDC operating under the auspices of the UN - World Meteorological Organization (WMO)*

We have included this information into the revision.

*40 Update statistics from current GRDC catalogue*

The revision now has updated statistics using the latest information from the GRDC (December 05 2017, which is available at ftp://ftp.bafg.de/pub/REFERATE/GRDC/website/grdc_summary_statistics.pdf).

*95 ARCTICNET is a now static database that is mirrored in the GRDC. In GRDC, stations of ARCTICNET are updated based on data deliveries to GRDC. The ARCTICHYCOS river discharge network is hosted and operated by GRDC with currently over 500 stations that are online available.*

We thank the reviewer for additional information about the status of ARC-TICNET, which we have included in the revision to provide a better overview about this data source.

Also we can see that the "GRDC" abbreviation used to represent the database we obtained from the Global Runoff Data Centre might create some confusion. We have carefully checked the Global Runoff Data Centre documentation and determined that "GRDB" is the more precise abbreviation to represent the database of 6,313 time series we obtained from the Global Runoff Data Centre (this abbreviation (GRDB) stands for "Global Runoff Data Base", as described at http://www.bafg.de/GRDC/EN/01_GRDC/13_dtbse/database_node.html). As a result, we have revised the manuscript (including tables, figures) to avoid confusion for users regarding to this database (i.e. GRDB now represents 6,313 stations of the Global Runoff Data Base that we obtained from the Global Runoff Data Centre). With the same intent, we reserve GRDC to refer only to the institution of "the Global Runoff Data Centre". The metadata file (GSIM_metadata.csv) and readme file of the database have also been adjusted to reflect this update.

Based on the most recent download of the GRDB, we suspect that, at present, the ARCTICNET data portal has not been fully integrated into GRDB since there are numerous stations available from ARCTICNET that do not appear in the GRDB (as shown in Figure 1). For this reason we treat ARCTICNET as an independent data source for GSIM but have commented in the manuscript that its future status is likely to be as a part of the GRDB.

*115 Most data of the European Water Archive (EWA) hosted by GRDC are available under the GRDC data policy and are no longer restricted to the FRIEND data policy. Based on data deliveries, the EWA is updated.*

Thank you for this information about the EWA data-policy and we have included it in the revision. Considering data availability, to our knowledge the EWA has not been fully integrated into GRDB. Figure 2 also demonstrates that there are numerous stations available from EWA that do not appear in the GRDB (e.g. in Spain, Italy, and Norway), which is consistent to the result of the de-duplication process in GSIM production (only 781 cases of duplication were detected). Thus we treat EWA and GRDB as two independent data sources for GSIM but have noted the status of EWA as a database hosted by GRDC in the manuscript.

*140 ARCTICNET is superseded by the arctic river basin database and ARCTICHY-COS databases that are part of the GRDC database. However, the ARCTICHYCOS database is operated as a project and data are open.*

Please see above two responses on this topic. We cannot establish from the latest GRDB dataset that all ARCTICNET gauges have been assimilated (Figure 1).

*230 It needs to be noted that data deliveries from national official data suppliers also contain errors. GRDC is performing plausibility checks on these data sets to detect and correct errors and provide feed-back to data suppliers.*

We have revised the manuscript to note that the Global Runoff Data Centre also performs quality control procedures to detect and correct errors in supplied data.

*Question: Have there been some checks to detect consistency of data sets supplied by national suppliers with data sets from the same stations contained in the GRDC? It is always necessary to check for the latest available versions of databases!*

In the initial stage of GSIM, we also made some checks to compare the temporal coverage of GRDB database (obtained in September 2016) and corresponding national suppliers. Generally, timeseries obtained from national suppliers representing the latest version of national streamflow databases as they were downloaded from national data portals, which have been updated regularly by national water agencies.

Table 1 provides a comparison of the first/last year of data entry between time series obtained from the Global Runoff Data Centre and national suppliers in corresponding countries. As mentioned in our manuscript, the number of data in national databases are much higher than the number of stations contained in the GRDB. In addition, many national databases have better coverage in time except for the average length of daily data in Australia, Brazil, Canada and the US. We anticipate this is likely due to strict selection criteria of these national data suppliers when transmitting their data to the Global Runoff Data Centre (i.e. only selecting data with longer periods coverage). Resulting from this information, we have decided to use national databases in preference to GRDB where available and have also included the caveat in the manuscript to ensure GSIM users are fully aware of this procedure.

*280 As standardization issues are a prominent issue it's important to describe if the development of the metadata catalogue has followed standards set (and endorsed by WMO as a standard setting organization), by the Open Geospatial Consortium (OGC), using WATER ML-2*
*320 Is GSIM Metadata compliant with OGC standards?*

We thank the reviewer for their comment on WATER ML-2 as endorsed by WMO. During development, the structure of GSIM metadata catalogue was mainly inspired by the Global Runoff Data Centre's data products (e.g.

GRDC catalogue), which have also been used as the standard for the development of a previous data product of our co-authors [4]. The WATER ML-2 is a comprehensive document regarding a standardised structure for the programming and documentation of online databases, as typically developed by national agencies. Providing this level of functionality is well beyond the capacity of our team, so that GSIM was not developed following this standard. Nonetheless, we consider that there are several opportunities for aligning our terminology with that specified by WATER ML-2 and we have endeavoured to match this aspect of WATER ML-2 as best as possible given our constraints.

[4] Gudmundsson, L. and Seneviratne, S. I.: Observational gridded runoff estimates for Europe (E-RUN version 1.0), Earth Syst. Sci. Data Discuss., 2016, 1-27, 2016.

*435 ff Provide information on the status of time series. In the case of archived time-series: describe whether there are update mechanisms in place or whether some of the data are closed data sets. There is the danger to generate orphaned data sets with incomplete information (metadata) on the version and last date of such data sets. This has often created confusion as researchers worked with outdated data sets (such as the UNESCO RivDis that still is used although it is outdated since over 20 years. It is used as the data is open without restrictions but the data holdings contain errors and/or have long since been replaced or updated including error correction.*

We thank the reviewer for the advice regarding to the status of original databases, which we also agreed is extremely useful to GSIM users. We have updated section 5.1 and Table 1 in the revision to mention this important information. We also added a cautionary sentence to ensure GSIM users are fully aware of possible errors of "static" stations. To avoid the danger of creating such an orphaned dataset, we have also followed the

GRDB metadata structure and included the first and last year of data entry corresponding to each station in the metadata (as described in section 4.1, step 3).

*510 Explain in a more transparent manner that GSIM will provide the metadata archive and not the actual time series as a result of different data policies from database operators including national services.*

We have included a section at the beginning of section 5 to emphasise that original time series cannot be made available, and thus the metadata catalogue has been developed to address this limitation. We also add some clarification in the Introduction section to ensure that data availability is discussed in a transparent manner.

*Discuss in more detail existing update mechanisms of databases and an indication which data sets are closed historic archives and which are living databases that are continuously updated.*

Regarding to the update mechanisms of sourced databases, we have provided more information in section 5 (as discussed above). We also revised Table 1 to clearly indicate which sources are closed historic databases and which sources are still being updated by data providers.

**Corrections in tables:**

*Table 1 ARCTICNET is part of GRDC, in addition, GRDC hosts the ARCTIC-HYCOS database; ARCTICNET is a closed historic database.*

We have updated Table 1 to clarify this information. We also clarified that (1) CHDP and GAME databases are also closed historic databases, (2)

EWA has been frozen since October 2014, and (3) the other databases are being updated by the data authority.

*Table 2 Example for Spain: EWA has 239 stations, GRDC 0 BUT: EWA is hosted under GRDC and data are available under GRDC data policy. These are no longer separated data bases!*

As discussed in previous comments, the terminology GRDC is now no longer used to indicate data source for GSIM, and thus this terminology has been replaced in Table 2 as "GRDB". However, we still keep these databases in two separate column to indicate that EWA was not fully integrated into GRDB (see the fifth comments).

[Figure]

**Figure 1.** Available stations in Russia. Red dots represent ARCTICNET database (139 stations). Blue dots represents GRDB (the Global Runoff Data Base) stations with daily record greater than 10 years (102 stations). GRDB stations were plotted on top of ARCTICNET stations.

**Fig. 1.**

[Figure]

**Figure 2.** Availability of GRDB and EWA databases. Red dots represent EWA database (3,731 stations). Blue dots represents GRDB stations with daily record greater than 10 years (3,104 stations). GRDB stations were plotted on top of EWA stations.

**Fig. 2.**

**Table 1.** Comparison of data availability between GRDB database and national databases.

| Seq | Country | National database | Data availability (GRDB) | | | | Data availability (national database) | | | |
|---|---|---|---|---|---|---|---|---|---|---|
| | | | No. of station | Earliest | Latest | Average length | No. of station | Earliest | Latest | Average length |
| 1 | Australia | bom | 358 | 1886 | 2012 | 47.24 | 2,941 | 1886 | 2016 | 31.25 |
| 2 | Brazil | ana | 439 | 1910 | 2010 | 36.48 | 3,313 | 1901 | 2016 | 29.29 |
| 3 | Canada | hydat | 1,029 | 1860 | 2014 | 45.84 | 6,325 | 1860 | 2015 | 26.97 |
| 4 | India | wris | 0 | NA | NA | NA | 318 | 1964 | 2015 | 30.03 |
| 5 | Japan | mlit | 151 | 1978 | 2003 | 12.42 | 1,029 | 1938 | 2014 | 22.78 |
| 6 | Spain | afd | 87 | 1977 | 1984 | 7.93 | 1,197 | 1912 | 2011 | 37.39 |
| 7 | US | usgs | 981 | 1873 | 2015 | 77.88 | 9,404 | 1880 | 2016 | 53.77 |

**Fig. 3.**